# Predictors of Life Satisfaction in Native Hawaiian and Pacific Islander College Students

**DOI:** 10.3390/ijerph22020164

**Published:** 2025-01-26

**Authors:** Kate McLellan, Narantsatsral Ganzorigta, Khulan Davaakhuu, Nicolas Spencer

**Affiliations:** Faculty of Sciences, Brigham Young University-Hawaii, Laie, HI 96762, USA; tsatsa@go.byuh.edu (N.G.); khulan@go.byuh.edu (K.D.); nspencer96@go.byuh.edu (N.S.)

**Keywords:** life satisfaction, Native Hawaiian, Pacific Islander, subjective well-being, college students

## Abstract

This study examines predictors of life satisfaction in Native Hawaiian and Pacific Islander (NH/PI) college students, who form an underrepresented group in well-being research. In a sample of 128 NH/PI students from a public university in Hawaii, self-report measures of life satisfaction, affect, stress, and mood were analyzed. The results indicate that positive affect and mood are significant positive predictors for life satisfaction, while negative affect predicts lower satisfaction. By contrast with other racial/ethnic groups, perceived stress did not significantly impact life satisfaction in NH/PI students. These findings suggest that emotional well-being is a key determinant for life satisfaction in NH/PI students, and stress is not. Culturally responsive interventions that focus on positive emotions and community support can enhance well-being and academic success in this population. This study contributes to the understanding of unique cultural influences on well-being in NH/PI students and provides implications for targeted interventions.

## 1. Introduction

Life satisfaction forms a crucial element in academic performance, as students who have high levels of life satisfaction tend to exhibit greater flexibility and resilience in the face of academic challenges [1]. Previous studies have found a strong relationship between life satisfaction, self-esteem, and academic performance among university students [1,2,3]. In particular, students who have higher life satisfaction are more invested in their educational experience, have greater confidence in their academic abilities, experience lower levels of academic stress, and are more positively motivated, all of which ultimately leads to higher GPAs [4].

Polynesians are perceived to have joyful communities, which is often associated with the robust cultural values and tightly knit social structures that are prevalent within Polynesian society. Familial bonds, which extend beyond immediate relationships to include extended family, often strengthen as members move away from home. This dynamic can foster a heightened respect for elders and a focus on communal well-being—values deeply embedded in the culture—that may enhance a collective sense of contentment and gratitude [5,6]. In addition, their profound affinity with nature, particularly the ocean, may further enrich their emotional well-being [7]. Polynesians exhibit a longstanding heritage of musical expression, dance, and oral traditions, which are pivotal elements for fostering a culture of exuberance and celebration. Polynesians have demonstrated resilience and an optimistic demeanor amid adversity [8].

Native Hawaiians exhibit a profound connection to their heritage, land, and spirituality, which has shaped their reputation for happiness. The essence of “aloha” forms the cornerstone of Hawaiian ethos, emphasizing love, empathy, and respect in interpersonal relations [9]. This ethos is manifest in the robust communal and familial ties that are prevalent in Hawaiian society, and, as such, their engagement with Hawaiian culture and the significance placed on Hawaiian identity correlate with lower stress levels and improved well-being [10]. Moreover, the rich heritage of hula, music, and storytelling is an integral conduit for cultural expression and may augment feelings of joy and communal belonging [10]. The captivating natural beauty of the Hawaiian Islands, characterized by their warm climates, stunning beaches, and lush landscapes, may further underpin a sense of contentment and subjective well-being (SWB), as seen in studies reporting associations between time in nature and improved cognitive function, mental health, physical activity, blood pressure, and sleep in children [11] and adults [11,12]. Recognizing that SWB forms a multifaceted experience, it is notable that Native Hawaiians exhibit remarkable resilience and a positive outlook in the face of adversity, supported by internal strengths such as hope, life satisfaction, and environmental mastery, as well as external resources, including social support and a strong identification with Native Hawaiian cultural values [13,14].

Despite these cultural strengths, Native Hawaiian and Pacific Islander (NH/PI) populations tend to be underrepresented in the study of well-being, as it relates to higher education. This study addresses this gap by investigating predictors of life satisfaction in NH/PI college students.

### 1.1. Subjective Well-Being

SWB represents an individual’s self-evaluation of their overall quality of life and emotional state, shaped by the personal assessments of thoughts, feelings, and circumstances. This concept incorporates the two key components of life satisfaction and emotional affect [15]. Life satisfaction refers to the cognitive judgment of how satisfied an individual is with their life, while affect involves emotional experiences, whether positive or negative [16].

### 1.2. Affect

Affect refers to the subjective experience and expression of emotions and moods. This includes feelings such as happiness, sadness, anger, and fear, as well as the physiological and behavioral changes that can accompany those feelings. Affective states can be fleeting or more long lasting, and they can be influenced by a variety of factors, including environmental, social, and personal factors. Affect is divided into positive and negative affect, both of which can have important implications for mental and physical health, as well as social and personal relationships [17].

Affect refers to the subjective experience and expression of emotions and moods, which includes feelings such as happiness, sadness, anger, and fear, together with associated physiological and behavioral changes. Influenced by various factors, including thinking, physical sensation, the environment, and social interactions [18], affect is categorized as positive or negative, and it has substantial implications for mental and physical health, as well as for interpersonal relationships [17].

Positive affect, including happiness and joy, exhibits a strong correlation with higher life satisfaction, and negative affect is associated with lower levels of SWB. The dynamic interplay among affect and life satisfaction exhibits a reciprocal relationship, in which positive affect enhances life satisfaction and vice versa [19]. The intricate relationship that holds between affect and SWB takes diverse factors into account, including personality, social support, and life circumstances.

### 1.3. Mood

Similar to affect, mood involves a prolonged emotional state that influences thinking, feeling, and behavior. Mood is characterized as positive (e.g., happy, content) or negative (e.g., sad, anxious), and it is subject to influences such as stress, hormonal changes, and environmental factors [20]. While the overall mood significantly impacts SWB, it is crucial to recognize that mood is only one factor, with others, such as social support, personal values, and life circumstances, playing substantial roles [21].

### 1.4. Stress

College students face an elevated risk of mental health challenges, including anxiety, depression, and stress. Academic and financial pressures, along with the impact of technological changes and social media, contribute to these challenges [22,23]. Previous research that links college students’ affect and mental health emphasizes correlations between SWB and daily affect [24].

The bidirectional relationship between environmental stressors and SWB underscores the influence of individuals’ perceptions of stress on their SWB [25]. Primary and secondary appraisals play crucial roles here, with the latter affecting SWB through an influence on the perceived adequacy of coping resources in response to stressors. Students’ interpretations of academic pressures and their capacity to address these demands can significantly shape their SWB. Previous studies indicate that academic stressors, such as performance pressures, heavy workloads, and time constraints, have a negative impact on students’ psychological well-being [26].

### 1.5. Justification and Purpose

Previous studies of college students’ SWB have predominantly encompassed non-Hispanic white/Caucasian, black/African American, Hispanic, and Asian populations, neglecting Native Hawaiian (NH) students and students from Pacific Island (PI) countries such as Samoa, Tonga, French Polynesia, New Zealand, Cook Islands, and Fiji. NH/PI high school students living in the United States graduate university at the same rate as the general population, but only 29% of NH/PIs aged 18–24 years were enrolled in a college or university, as opposed to 57% of Asians and 39% of non-Hispanic white adults [27,28]. A qualitative study of PI high school and college-aged adults identified some key disparities in access to educational resources, socioeconomic resources, stereotyping, and cultural differences that could impede PI students’ ability to succeed in college or university settings [28,29].

Native Hawaiians and Pacific Islanders face unique cultural, socioeconomic, and environmental challenges that could influence their SWB, including strong familial and community ties, connection to their ancestral lands, and exposure to colonialism and minority status [30,31]. It is crucial to center the voices and experiences of NH/PI college students to understand the unique factors that contribute to their SWB. Doing so can inform tailored interventions and support services for the promotion of holistic well-being and academic success.

This study investigates the following questions: (1) what are the primary predictors for life satisfaction among NH and Pacific Islander college students? and (2) how do these predictors differ, if at all, from those that are observed in other racial/ethnic groups of college students?

## 2. Materials and Methods

This quantitative study aimed to examine the predictors for life satisfaction among NH and Pacific Islander college students. It was approved by Brigham Young University–Hawaii’s Institutional Review Board (approval #23-04) and conducted in adherence with the 1964 Helsinki Declaration and 45 CFR 46 of Federal Policy for the Protection of Human Subjects, as well as its later amendments or comparable ethical standards. The participants were undergraduate students identifying as Native Hawaiians or Pacific Islanders, recruited from a small public university in Hawaii. The participants completed an online survey collecting demographic and emotional health measures and assessing life satisfaction, positive and negative affect, mood, and perceived stress, using Qualtrics to collect responses (Qualtrics LLC, Provo, UT, USA). Informed consent was obtained from all participants before they completed the questionnaire. All data were collected from 15 January to 15 February 2023.

### 2.1. Instruments

#### 2.1.1. Satisfaction with Life Scale (SWLS)

Life satisfaction was measured using the SWLS, which functions as an assessment tool for the appraisal of an individual’s overall life satisfaction [32]. Made up of five statements exploring diverse facets of life contentment, for example, “The conditions of my life are excellent” and “If I could live my life over, I would change almost nothing”, this instrument collects respondents’ agreement on a seven-point scale from “strongly agree” to “strongly disagree”, with higher cumulative scores, ranging from 5 to 35, reflecting greater satisfaction with life. Deiner et al. [32] reported the Cronbach alpha reliability of 0.78.

#### 2.1.2. Positive and Negative Affect Scale (PANAS)

The PANAS was employed to evaluate the participants’ positive and negative affect [18]. This is a validated psychological instrument that is used for gauging an individual’s spectrum of positive and negative emotions (α = 0.80 for the full scale) and is widely used in cross-cultural studies [33]. The PANAS assesses positive affect (PA) with 10 positive emotion-related items such as “interested”, “enthusiastic”, and “proud”, with high levels associated with enthusiasm and vitality, and low levels linked to sadness and lethargy.

Negative Affect (NA) is evaluated through 10 negative emotion-related items such as “distressed” and “ashamed” (α = 0.85). Elevated NA scores signify heightened distress, whereas low levels indicate a state of calm and tranquility. Respondents rate the intensity of experienced emotions over a defined period (e.g., the previous week) on a five-point scale. PA and NA scores range from 0 to 50, with higher scores indicating heightened emotional intensity.

The Positive Affect Scale of the PANAS demonstrated high internal consistency, with Cronbach’s alpha ranging from 0.86 to 0.90. The Negative Affect Scale also showed strong internal consistency, with Cronbach’s alpha between 0.84 and 0.87. Test–retest reliability over an 8-week period yielded correlations of 0.47 to 0.68 for Positive Affect and 0.39 to 0.71 for Negative Affect. The PANAS has demonstrated strong validity correlations with measures of general distress and dysfunction, depression, and state anxiety [34].

#### 2.1.3. Perceived Stress Scale (PSS)

Perceived stress was measured using the Perceived Stress Scale (PSS), a self-report scale intended to assess an individual’s perception of stress in their life [35]. The 10-item instrument evaluates the degree to which individuals appraise their life circumstances as stressful, unpredictable, and beyond their control (α = 0.78). The respondents indicated how often they had felt or thought a certain way in the previous month on a 5-point scale; items included, for example, “In the last month, how often have you felt nervous and stressed?”, “In the last month, how often have you felt that things were going your way?”, and “In the last month, how often have you been angry because of things that happened that were outside of your control?”. Using the scoring algorithm to reverse-score the responses to four of the questions, we summed scores ranged from 0 to 40, with higher scores suggesting higher perceived stress [35].

#### 2.1.4. Brief Mood Inventory Scale (BMIS)

The participants’ prevailing mood states, in terms of positive and negative aspects, were measured using the BMIS [36]. Consisting of statements encompassing positive and negative moods (PM and NM, respectively) such as “lively”, “tired”, “jittery”, and “active”, the BMIS presents eight descriptors each for the two mood dimensions. Respondents evaluate their agreement with each statement on a five-point scale. The scores for PM and NM ranged from 8 to 40, indicating the intensity of the mood states measured [36]. Wang et al. reported good cross-cultural adaptability of the BMIS when comparing American and Chinese college students [37] and Visoiu et al. determined Cronbach’s alpha to be 0.80 [38].

### 2.2. Data Analysis

The data were analyzed using multiple linear regression to identify the predictors for life satisfaction in NH and Pacific Islander college students. Additional analyses examined group differences in the predictors for life satisfaction between NH/PI students and other racial/ethnic groups. Analyses were conducted using Jamovi version 2.3 (Jamovi, Sydney, Australia) [39]. Pearson’s correlation was used to examine the relationships between measures. Multiple and forward regression models were used to predict the factors in higher life satisfaction scores. The respondents’ sex, age, year in school, student generation, credit load, and GPA were controlled in the regression models. Significance was set at p < 0.05.

## 3. Results

The sample included 128 NH and Pacific Islander undergraduate students, 64 men and 64 women, with a mean age of 24.1 years (SD = 6.2). In this group, 53.1% (68/128) held US citizenship (domestic), while 46.8% (60/128) held citizenship in a Pacific Island country (international); 35% (32/60) identified as first-generation college students, and 97.6% (124 of 128) were full-time students enrolled for ≥12 credits. (See additional subject demographics in Table 1.)

Pearson correlations were computed for all outcome measures and are summarized in Table 2. Notably, PA showed a significant negative correlation with NA (r = −0.467, *p* < 0.001), and positive correlations were observed between PA and PM (r = 0.711, *p* < 0.001), as well as PA and SWLS (r = 0.67, *p* < 0.001). SWLS also demonstrated a significant positive correlation with PM (r = 0.624, *p* < 0.001). PSS, however, was only significantly correlated with NM (r = 0.538, *p* < 0.001). These findings indicate a positive multi-directional relationship among PA, PM, and SWLS scores, suggesting that an increase in one scale may coincide with elevations in the other two variables. Conversely, PSS only showed a significant correlation solely with NM (r = 5.38, *p* < 0.001) (Table 2).

A forward-model regression analysis was conducted to evaluate the predictors of life satisfaction among NH/PI students. The overall model was statistically significant (F = 29.47, *p* < 0.001), and it accounted for 52.4% of the variance in SWLS scores. PA (β = 0.54, *p* < 0.001) and PM (β = 0.64, *p* < 0.048) were significant positive predictors, and NA was a significant negative predictor of life satisfaction (β = −0.37, *p* < 0.33) (Table 3).

Notably, perceived stress did not contribute significantly to the model, which suggests that the emotional and affective components may be stronger determinants for life satisfaction in this population than in the appraisal of cognitive stress.

## 4. Discussion

This study provides novel insights into the predictors of life satisfaction in NH and Pacific Islander college students, a population that has been historically underrepresented in the relevant literature. The findings of this study highlight the central role that positive and negative affect play in determining life satisfaction and the lack of such a role for perceived stress, which was not a significant predictor in this sample.

This result is in alignment with previous research that suggests that affective factors could be more salient than cognitive stress appraisals in shaping SWB, particularly for marginalized groups [30,40,41]. The emphasis on positive emotions and mood states as key predictors in life satisfaction suggests that interventions that are aimed at promoting PA and well-being could be particularly beneficial for the enhancement of life satisfaction in NH/PI students. It should be noted that the regression model accounted for over 50% of the variance in life satisfaction, which indicates that the identified predictors are robust and meaningful for capturing the determinants of well-being in this population. These findings contribute to the limited empirical literature on the psychosocial factors that are associated with the life satisfaction of NH and Pacific Islander college students, and they have implications for ways to support educational success and retention in this underserved group.

In a comparison between the regression findings of this study to the results of previous studies on other racial/ethnic groups of college students, some noteworthy differences emerged. Perceived stress was found to be a significant negative predictor for life satisfaction in Asian, Black, and Latino students [30,40], but it did not predict life satisfaction in NH/PI students in this study. This may indicate cultural differences in the recognition of and coping with stress, as well as in its relative influence on overall life satisfaction. For example, NH/PI cultures tend to emphasize collectivist values, community support, and spirituality, which may buffer the negative impact of stress on well-being [1,30,42].

Consistent with earlier research on other racial/ethnic groups, PA and positive mood were significant positive predictors of life satisfaction for NH/PI students. This suggests that the cultivation of positive emotions and experiences may be an important pathway for enhancing overall life satisfaction in this population [1]. Researchers examining life satisfaction among Turkish college students found that major satisfaction, self-efficacy, and academic performance were the key predictors [43]. Similarly, a study on predictors of life satisfaction in New Zealand found that demographic factors, as well as subjective experiences like health status and social trust, were significant predictors [44]. Studies have found that factors such as major satisfaction, self-efficacy, academic performance, and demographic characteristics like age, gender, and marital status can all significantly influence life satisfaction for college students [45].

A similar study conducted with Peruvian college students examining emotional intelligence, self-esteem, and resilience reported a similar mean score on the SWLS of 23.5 (SD = 6.5) compared to 24.7 (SD = 6.2) in the present study. Vilca-Pareja et al. found that all measures examined had significant and positive correlations with SWLS and highlighted the need to create emotional education programs for young adults to increase their life satisfaction [46].

## 5. Conclusions

This study provides important insights into the psychosocial determinants of life satisfaction for NH and Pacific Islander college students and highlights the central role of positive and negative affect. These findings have important implications for the development of targeted interventions and support services to promote well-being and academic success in this underrepresented student population. Further research is required to replicate and expand upon these findings and to identify other potential predictors of life satisfaction, including factors related to cultural identity, social support, and academic success.

### Implications and Future Considerations

This study provides a preliminary understanding of key predictors of life satisfaction in NH/PI students. Its findings indicate the central role that positive emotional experiences, rather than cognitive stress appraisal, have in determining life satisfaction in this population. Recognition of these unique determinants can inform culturally responsive interventions to promote overall well-being in NH/PI college students.

The limitations of this study include its cross-sectional design, which precludes the making of causal inferences, and the reliance on self-report measures. Future research should explore the role that cultural factors, such as indigenous values, traditions, and support systems, play in shaping the predictors of life satisfaction in NH/PI college students.

## Figures and Tables

**Table 1 ijerph-22-00164-t001:** Sociodemographic characteristics of the participants.

Participants		N	%
Age (years)		24.1 ± 6.2	
Sex			
	Male	64	50
	Female	64	50
Year in School			
	First year	39	30.7
	Second year	31	24.4
	Third year	27	21.3
	Fourth year	30	23.6
Generation of student			
	First generation	44	34.3
	Second generation	53	41.4
	Third generation	27	21.1
	Fourth generation	4	3.1
GPA			
	Low (<2.99)	34	26.5
	Moderate (3.0–3.59)	46	35.9
	High (3.6–4.0)	48	37.5
Total		128	100

**Table 2 ijerph-22-00164-t002:** Comparative analysis of means and correlations between affect, mood, life satisfaction, and perceived stress.

Variables	Mean (SD)	PA	NA	PM	NM	SWLS	PSS
(PA)	32.5 (6.8)	-					
(NA)	23.3 (7.2)	−0.467 ***	-				
(PM)	20.1 (3.6)	0.711 ***	−0.182	-			
(NM)	22.5 (4.6)	0.053	0.200	0.372	-		
(SWLS)	24.7 (6.2)	0.670 ***	−0.371 *	0.624 ***	0.152	-	
(PSS)	22.0 (4.1)	−0.168	0.283	0.098	0.538 ***	−0.005	-

Note. * significant at the 0.05 level of significance, *** significant at the 0.001 level of significance. SD = standard deviation, PA = Positive Affect, NA = Negative Affect, PM = Positive Mood, NM = Negative Mood, SWLS = Satisfaction with Life Scale, PSS = Perceived Stress Scale.

**Table 3 ijerph-22-00164-t003:** Forward model regression predicting satisfaction with life.

Variable	*β*	*p*	SE	95% CI
PA	0.54 ***	<0.001	0.10	(0.33, 0.73)
PM	0.23 *	0.017	0.098	(0.04, 0.42)
NA	−0.37 *	0.033	0.161	(−0.71, −0.05)
NM	−0.20	0.21	0.265	(−0.34, 0.75)
PSS	−0.01	0.94	0.15	(−0.3, 0.33)

Note. Dependent Variable: SWLS. R^2^ = 0.524, F = 29.47, *p* < 0.001. * Significant at the 0.05 level of significance, *** significant at the 0.001 level of significance. β = Standardized beta coefficient, SE = Standard Error, CI = confidence interval. PA = Positive Affect, NA = Negative Affect, PM = Positive Mood, NM = Negative Mood, PSS = Perceived Stress Scale.

## Data Availability

The datasets presented in this article are not readily available because they are part of an ongoing study. Requests to access the datasets should be directed to Kate McLellan at kate.mclellan@byuh.edu.

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
