# Peer review of "Predictors of Life Satisfaction in Native Hawaiian and Pacific Islander College Students"

_ijerph, 2025, doi:10.3390/ijerph22020164_

Round 1

Reviewer 1 Report

Comments and Suggestions for Authors

Dear authors,

Congratulations on your excellent work presented on Predictors of Life Satisfaction in Native Hawaiian and Pacific 2 Islander College Students.

As for recommendations to improve it, I recommend updating greatest scientific contributions and more references from the last five years on the subject of study, especially in the Introduction and Discussion sections.

In addition, I suggest that you provide the DOI numbers in the References section, since although they are not mandatory, they are highly recommended to be able to cite them.

Author Response

1. Summary

2. Questions for General Evaluation

Reviewer’s Evaluation

Response and Revisions

Does the introduction provide sufficient background and include all relevant references?

Can be improved

We have updated the references and included additional background information to explain the subject of the study and added DOI numbers, as requested.

Are all the cited references relevant to the research?

Yes

While this was not a criterion on the Review Report form, we did go through the references to verify they’re all relevant and cited in the study.

Is the research design appropriate?

Can be improved

We expanded the Discussion section to clarify the strength and limitations of the study and how future research could prove even more valuable.

Are the methods adequately described?

Can be improved

We expanded the Methods section to explain our study timeline and procedures.

Are the results clearly presented?

Yes

Are the conclusions supported by the results?

Yes

3. Point-by-point response to Comments and Suggestions for Authors

Comments 1: Congratulations on your excellent work presented on Predictors of Life Satisfaction in Native Hawaiian and Pacific Islander College Students.

As for recommendations to improve it, I recommend updating greatest scientific contributions and more references from the last five years on the subject of study, especially in the Introduction and Discussion sections. 

Response 1: Thank you for pointing this out. We agree with this comment. Therefore, we have included additional references from the last 5 years to bring more recent and relevant credibility to support our statements.

Comments 2: In addition, I suggest that you provide the DOI numbers in the References section, since although they are not mandatory, they are highly recommended to be able to cite them.

Response 2: Thank you for pointing this out. We have, accordingly, added DOI numbers to all references to allow reviewers and readers to research the linked citations. done/revised/changed/modified…..to emphasize this point.

4. Response to Comments on the Quality of English Language

Point 1: “The quality of English does not limit my understanding of the research.”

Response 1: Thank you

Reviewer 2 Report

Comments and Suggestions for Authors

The article presents research on life satisfaction and variables that may explain the variance of this parameter among Native Hawaiian and Pacific Islander college students, an underrepresented group in well-being research. 

Overall, the article is well organized and clearly written. It lacks the presentation of Cronbach's Alpha value, which indicates the internal consistency of the items of each scale, and this cannot be left as it is. 

Since some notable differences emerged when comparing the regression results with previous studies on other groups, it would be of interest to mention more explicitly the differences, perhaps with some values of means and standard deviations in other geographically close ethnic groups of similar ages. 

Author Response

Response to Reviewer 2 Comments

1. Summary

2. Questions for General Evaluation

Reviewer’s Evaluation

Response and Revisions

Does the introduction provide sufficient background and include all relevant references?

Yes

Thank you

Are all the cited references relevant to the research?

Yes

Thank you

Is the research design appropriate?

Yes

Thank you

Are the methods adequately described?

Can be improved

We made the suggested edits to the statistical tests and included comparisons with other ethnic groups

Are the results clearly presented?

Yes

Are the conclusions supported by the results?

Yes

3. Point-by-point response to Comments and Suggestions for Authors

Comments 1: It lacks the presentation of Cronbach's Alpha value, which indicates the internal consistency of the items of each scale, and this cannot be left as it is. 

Response 1: Thank you for pointing this out. We agree with this comment. Therefore, we have included the alpha value for each of the survey tools used in the study to assess affect, mood, stress, and life satisfaction. You will find these changes in section 2.1 Measures.

Comments 2: Since some notable differences emerged when comparing the regression results with previous studies on other groups, it would be of interest to mention more explicitly the differences, perhaps with some values of means and standard deviations in other geographically close ethnic groups of similar ages. 

Response 2: The data on other Pacific Island young adult communities is scarce, so we compared our outcomes with those of college aged Asian, Latino, African American students in the United States as well as students in China, Peru, and Turkey. There were many studies on life satisfaction predictors in older Polynesians, but many of the measures they examined (e.g. income, access to healthcare, and intergenerational relationships) were not included in the present study. Future studies could examine similar metrics, as they apply to young adults and university students.

4. Response to Comments on the Quality of English Language

Point 1: “The quality of English does not limit my understanding of the research.”

Response 1: Thank you

5. Additional clarifications

Reviewer 3 Report

Comments and Suggestions for Authors

The study addresses an important area. 

Below are my overall comments:

Abstract

I am convinced that the abstract covers all the essential academic elements contained in the full-length paper article .i.e.  background, purpose, materials and methods, measures, data analysis, results, conclusions, and implications.

The following suggestions may improve the abstract: Background (opening sentence or two placing the research in context. It should briefly tell the gap the study is filling - "what is known and why this study is necessary") Purpose (one sentence giving the purpose of the current work), Methods (one or two sentences explaining what was done i.e. study design, sampling techniques adopted, methods and instruments of data collection, and the method and instrument for data analysis),  Results (a sentence or two indicating the main findings, results, or outcomes of the research) Conclusions and implications (a sentence or two giving the most important consequence of the research study i.e. what do the results mean and what are the implications). 

The abstract should summarize the main aspects of the entire research article. It should help the reader understand the scope of the research paper. 

Introduction

The introduction omits the crucial element of the study (cultural dimension) and its relationship with life satisfaction (see line 49). Assertions are made with sources e.g. line 27, 29 - 35. The words "aloha"(line 39), "hula" (line 42) should be in italics. 

Materials and methods 

See comments on abstract above. Ethical considerations should be included in these section to demonstrate adherence to research ethics and integrity. 

Results, discussion, and conclusion

It is not convincing how the results, discussion, and conclusion pertain to the study aims.

Author Response

Response to Reviewer 3 Comments

1. Summary

2. Questions for General Evaluation

Reviewer’s Evaluation

Response and Revisions

Does the introduction provide sufficient background and include all relevant references?

Must be improved

Thank you

Is the research design appropriate?

Can be improved

We expanded the Discussion section to clarify the strength and limitations of the study and how future research could prove even more valuable.

Are the methods adequately described?

Must be improved

Are the results clearly presented?

Must be improved

Are the conclusions supported by the results?

Must be improved

3. Point-by-point response to Comments and Suggestions for Authors

Comments 1: The abstract should summarize the main aspects of the entire research article. It should help the reader understand the scope of the research paper. 

Response 1: Thank you for pointing this out. We have edited the abstract to ensure it is following the publisher’s formatting and is structured with 1-2 sentences each for the background, methods, results, and conclusions. Please see page 1, lines 7-17.

Comments 2: The introduction omits the crucial element of the study (cultural dimension) and its relationship with life satisfaction (see line 49). Assertions are made with sources e.g. line 27, 29 - 35. The words "aloha"(line 39), "hula" (line 42) should be in italics. 

Response 2: Agree. I/We have, accordingly, done/revised/changed/modified…..to emphasize this point. Discuss the changes made, providing the necessary explanation/clarification. Mention exactly where in the revised manuscript this change can be found – page number, paragraph, and line.]

“[updated text in the manuscript if necessary]”

Comments 3: Materials and methods: Ethical considerations should be included in these section to demonstrate adherence to research ethics and integrity. 

Response 3: We edited the wording to make the statement about institutional review board approval for the study and obtaining an informed consent from all participants prior to them completing the study questionnaire. Please see page 3, section 2. Materials and Methods, lines 127-138. Updated text:

“This quantitative study aimed to examine the predictors of life satisfaction among Native Hawaiian and Pacific Islander college students and was approved by the university’s Institutional Review Board (approval #23-04) and conducted in adherence with the 1964 Helsinki Declaration, 45 CFR 46 of Federal Policy for the Protection of Human Subjects and, and its later amendments or comparable ethical standards. Informed consent was obtained from all participants before completing the questionnaire.”

Comments 4: Results, discussion, and conclusion:

It is not convincing how the results, discussion, and conclusion pertain to the study aims.

Response 4: Agree. I/We have, accordingly, done/revised/changed/modified…..to emphasize this point. Discuss the changes made, providing the necessary explanation/clarification. Mention exactly where in the revised manuscript this change can be found – page number, paragraph, and line.]

“[updated text in the manuscript if necessary]”

4. Response to Comments on the Quality of English Language

Point 1: “The quality of English does not limit my understanding of the research.”

Response 1: Thank you

5. Additional clarifications

Round 2

Reviewer 3 Report

Comments and Suggestions for Authors

Far-reaching assertions are made (27-32) and (35 -39) without reference sources.

Author Response

Comment: Far-reaching assertions are made (27-32) and (35 -39) without reference sources.

Aurthor response: 

Thank you for your feedback on our manuscript, and we sincerely apologize for not addressing this issue in the previous edits. To ensure the Background section provides a more comprehensive and educational experience for the reader, we have added eight additional references and included clarifying statements. These additions incorporate relevant and timely material to strengthen the context and enhance the overall quality of the manuscript.

We greatly appreciate your guidance throughout this process and hope the revised content aligns with your expectations. Please let us know if there are any further adjustments required.

You will find the additions highlighted with author comments in lines 27-50 of the attached manuscript. 
